# Impact of inconsistent ethnicity recordings on estimates of inequality in child health and education data: a data linkage study of Child and Adolescent Mental Health Services in South London

Alice Wickersham [ID],[1] Jayati Das-Munshi [ID],[2] Tamsin Ford [ID],[3] Amelia Jewell [ID],[4] Robert Stewart,[2] Johnny Downs [ID] [1]

For numbered affiliations see end of article.

**Correspondence to**
Alice Wickersham;
alice.wickersham@kcl.ac.uk

## ABSTRACT

**Objectives** Ethnicity data are critical for identifying inequalities, but previous studies suggest that ethnicity is not consistently recorded between different administrative datasets. With researchers increasingly leveraging cross-domain data linkages, we investigated the completeness and consistency of ethnicity data in two linked health and education datasets.

**Design** Cohort study.

**Setting** South London and Maudsley NHS Foundation Trust deidentified electronic health records, accessed via Clinical Record Interactive Search (CRIS) and the National Pupil Database (NPD) (2007–2013).

**Participants** N=30 426 children and adolescents referred to local Child and Adolescent Mental Health Services.

**Primary and secondary outcome measures** Ethnicity data were compared between CRIS and the NPD. Associations between ethnicity as recorded from each source and key educational and clinical outcomes were explored with risk ratios.

**Results** Ethnicity data were available for 79.3% from the NPD, 87.0% from CRIS, 97.3% from either source and 69.0% from both sources. Among those who had ethnicity data from both, the two data sources agreed on 87.0% of aggregate ethnicity categorisations overall, but with high levels of disagreement in Mixed and Other ethnic groups. Strengths of associations between ethnicity, educational attainment and neurodevelopmental disorder varied according to which data source was used to code ethnicity. For example, as compared with White pupils, a significantly higher proportion of Asian pupils achieved expected educational attainment thresholds only if ethnicity was coded from the NPD (RR=1.46, 95% CI 1.29 to 1.64), not if ethnicity was coded from CRIS (RR=1.11, 0.98 to 1.26).

**Conclusions** Data linkage has the potential to minimise missing ethnicity data, and overlap in ethnicity categorisations between CRIS and the NPD was generally high. However, choosing which data source to primarily code ethnicity from can have implications for analyses of ethnicity, mental health and educational outcomes. Users of linked data should exercise caution in combining and comparing ethnicity between different data sources.

## STRENGTHS AND LIMITATIONS OF THIS STUDY

⇒ We leveraged a data linkage between routinely collected health and education datasets for a large cohort of children and adolescents referred to mental health services in South London.
⇒ We conducted a comprehensive analysis of the availability and consistency of ethnicity data between these health and education datasets.
⇒ Only one assessment of ethnicity was available from each data source and for each individual, precluding analysis of within-person changes in identity over time.
⇒ Aggregate ethnicity data were primarily used for analysis.

## INTRODUCTION

Ethnicity data are critical for highlighting inequalities in the population and informing public health and educational policies. For example, some minority ethnic groups are thought to perform less well in school assessments,[1–3] and to be underdiagnosed with neurodevelopmental disorders.[4] A recent review reported significant ethnic inequalities in access, experiences and outcomes of healthcare in the UK, but also highlighted that very few national datasets have sufficiently high-quality ethnicity data to robustly investigate these inequalities.[5] Incomplete and inconsistent ethnicity data in administrative datasets could result in variable estimates for important clinical and educational outcomes, compromising our ability to identify and address these inequalities.

Ethnicity is a fluid construct which should be self-ascribed and is socially constructed, capturing shared characteristics such as geography, ancestry, culture and language.[6] Epidemiological studies using routinely collected data often rely on ethnicity variables which

have been aggregated into several top-level categories. This is due to small sample sizes in more granular ethnic groups, associated concerns with statistical power and the risk of inadvertent deanonymisation which arises when sharing detailed ethnicity data. However, aggregating ethnicity data results in problematic within-group heterogeneity, and risks specific ethnic groups being overlooked in research.[6–8] Not only do these aggregate classifications fail to meaningfully capture homogenous groups, but they also do not necessarily align across different data sources. Previous research from the USA, Australia and the UK has repeatedly raised concerns over the completeness and consistency of ethnicity variables in routinely collected health and other administrative data, particularly in comparison to self-reported survey data.[9–14]

A recent UK report on Hospital Episode Statistics, the Emergency Care Data Set and the Community Services Dataset found that ethnicity recordings in these datasets were often incomplete, and that for minority ethnic groups in particular, ethnicity codes were not used consistently for the same individuals over time.[15] Another UK study compared two linked administrative health datasets from primary care and hospital settings and found that combining the data sources increased the completeness of ethnicity data.[16] While agreement between these sources was generally high, with 85% of patients having the same ethnicity recorded in both primary care and hospital data, agreement was much weaker for South Asian and Other ethnic groups.

We sought to explore this issue in linked administrative data from health and education domains for a cohort of pupils who had been referred to Child and Adolescent Mental Health Services (CAMHS) in South London. With cross-disciplinary data linkages increasingly being adopted for epidemiological research in this age group,[17] an understanding of the completeness and consistency of ethnicity from two such data sources would be beneficial. This will help us to understand whether and how estimates of ethnic inequalities in health and education services might be affected by any measurement issues specific to these datasets and to populations of children and adolescents accessing mental health services.

The aims of this exploratory analysis were therefore to (1) summarise the completeness of ethnicity data in the linked health and education records; (2) summarise the alignment of aggregate ethnicity between the two data sources; and (3) investigate whether key educational and clinical outcomes vary according to which data source is used to code ethnicity.

## MATERIALS AND METHODS
### Data
We used an existing data linkage between deidentified electronic health records from South London and Maudsley NHS Foundation Trust (SLaM), and educational records from the Department for Education. SLaM serves the boroughs of Croydon, Lambeth, Lewisham and Southwark, and provides some national services. The SLaM catchment area is ethnically diverse and has a lower proportion of individuals from White ethnic backgrounds compared with London and England as a whole.[18] SLaM makes deidentified electronic health records available for research via the Clinical Record Interactive Search (CRIS) system. For children and adolescents (aged 4–18 years) referred to SLaM CAMHS between 2007 and 2013, individual-level data linkage was undertaken between CRIS and the National Pupil Database (NPD).[19 20] This analysis focused on pupils in the CRIS-NPD data linkage who had been referred to SLaM CAMHS, either from within the SLaM catchment area or from out of area (n=30 426). Reporting follows the REporting of studies Conducted using Observational Routinely-collected health Data (RECORD) guidelines (online supplemental table S1 and figure S1).

The CRIS data resource, including the linked data used in this manuscript, has received research ethics approval for secondary analyses (Oxford REC C, reference 18/SC/0372). The CRIS Oversight Committee ensures that research conducted using health records is ethical and legal, and service users can opt-out of their CRIS data being used for research.

### Patient and public involvement statement
The data linkage used in this study was carried out in consultation with several patient and caregiver groups at the National Institute for Health Research Biomedical Research Centre, SLaM and King's College London. This has been described elsewhere.[19 20]

### Ethnicity—NPD
In the NPD, pupil ethnicity is collected as part of the school census undertaken by England's state schools. It is either self-ascribed or parent ascribed, although historically this information could also be ascribed by the school.[21] Due to concerns over sensitivity and potential disclosure, ethnicity extracts from the NPD are sometimes supplied to researchers as what is termed 'major ethnic groups' only (online supplemental table S2). Major ethnic groups are similar to the aggregated groups used by the Office for National Statistics; however, Chinese is provided as a separate group rather than aggregated under Asian/Asian British (White, Black, Asian, Chinese, Mixed, Other and Unknown).[22]

### Ethnicity—CRIS
In SLaM source data, accessed via CRIS, patient ethnicity is recorded in structured fields and intended to be self-ascribed, although in practice, ethnicity may sometimes be assumed and ascribed by the clinician or other staff member completing the record. A patient's ethnicity may be recorded during their interactions with any SLaM service (including, but not limited to, CAMHS). CRIS aggregates ethnicity recorded in structured fields into 16+1 categories (online supplemental table S3), and this is the format in which the ethnicity variable is

made available to researchers. For the pupils in this study, ethnicity was extracted from electronic health records via CRIS in March 2020.

To assist comparison with the NPD, we further aggregated CRIS ethnicity into White, Black, Asian, Chinese, Mixed, Other and Unknown (online supplemental table S4), although it should be noted that some inconsistencies between the resulting aggregate ethnic groups remain. Most notably, the NPD places Kurdish groups into the Other major ethnic group, while CRIS places them in the 16+1 category 'any other White background', such that they were subsequently aggregated as White in CRIS. Also, the NPD places Malaysian groups into the Chinese major ethnic group, while CRIS places them in the 16+1 category 'any other ethnic group', such that they were subsequently aggregated as Other in CRIS. Due to the aforementioned concerns around data security and confidentiality, we were unable to access more granular ethnicity variables to make these categorisations consistent between the data sources.

### Statistical analysis

First, we summarised the completeness of ethnicity variables in the NPD and CRIS. We initially did this by including the Unknown categories, then excluding these categories to determine the availability of what a previous study termed 'usable ethnicity'.[16] Second, we used descriptive statistics and Cohen's kappa to explore how usable ethnicity compared between the NPD and CRIS for those who had ethnicity variables available from both data sources.[23 24]

We produced risk ratios to estimate how ethnicity from each data source was associated with the outcome variables of neurodevelopmental disorder and year 11 educational attainment. Neurodevelopmental disorder was defined as any record of an intellectual disability (International Statistical Classification of Diseases and Related Health Problems—10th Revision (ICD-10) codes F70x–F79x), pervasive developmental disorder (ICD-10 codes F84x), or hyperkinetic disorder (ICD-10 codes F90x), derived from CRIS structured primary and secondary diagnosis fields.[25] We focused on neurodevelopmental disorders because they are often pervasive and lifelong,[26] therefore overcoming biases which might arise in differential timing of diagnoses. Year 11 educational attainment was derived from the NPD, and defined as achieving expected attainment thresholds of five A* to C General Certificates of Secondary Education or equivalent grades including English and maths. These assessments typically take place at ages 15–16 years. A reduced sample of n=14 567 had available year 11 attainment data and were included in these analyses, consistent with the fact that our cohort of pupils were age 4–18 at the time of linking their educational records, such that many had not yet completed their year 11 assessments by the time of the linkage window. In order to understand how missing ethnicity data in

**Table 1** Sample overview

| | N (%) |
|---|---|
| **Usable NPD-derived ethnicity, n=24 127** | |
| White | 13 259 (55.0) |
| Black | 6278 (26.0) |
| Asian | 755 (3.1) |
| Chinese | 73 (0.3) |
| Mixed | 3247 (13.5) |
| Other | 515 (2.1) |
| **Usable CRIS-derived ethnicity, n=26 473** | |
| White | 14 572 (55.0) |
| Black | 7400 (28.0) |
| Asian | 959 (3.6) |
| Chinese | 82 (0.3) |
| Mixed | 2294 (8.7) |
| Other | 1166 (4.4) |
| **Year 11 expected attainment threshold achieved, n=14 567** | |
| No | 10 009 (68.7) |
| Yes | 4558 (31.3) |
| **Neurodevelopmental disorder diagnosis, n=30 426** | |
| No | 21 856 (71.8) |
| Yes | 8570 (28.2) |

CRIS, Clinical Record Interactive Search; NPD, National Pupil Database.

either source might introduce bias into analyses of these variables, we combined all Unknown labels and empty cells into a single 'Missing' category.

Finally, in series of sensitivity analyses aiming to understand the impact of supplementing missing ethnicity data in one source using ethnicity data from the other source, we repeated these risk ratios, first using NPD ethnicity supplemented from CRIS when missing from the NPD, then using CRIS ethnicity supplemented from the NPD when missing from CRIS. All analyses were conducted in Stata V.15.1.[27]

## RESULTS
### Completeness of ethnicity variables

Including the Unknown categories, NPD ethnicity was available for n=25 951 (85.3%), and CRIS ethnicity was available for n=28 363 (93.2%). Excluding the Unknown categories, usable NPD ethnicity was available for n=24 127 (79.3%), and usable CRIS ethnicity was available for n=26 473 (87.0%) (table 1). In total, n=29 609 (97.3%) had usable ethnicity data from either the NPD or from CRIS, and n=20 991 (69.0%) had usable ethnicity data from both sources. Other sociodemographic information for the linked cohort has been reported elsewhere.[20]

**Table 2** Cross-tabulations of CRIS and NPD ethnicity where ethnicity was available from both sources

| | CRIS | | | | | |
| --- | --- | --- | --- | --- | --- | --- |
| | White n=11850 | Black n=5743 | Asian n=675 | Mixed n=1840 | Other n=808 | Total n=20916 |
| **NPD** | | | | | | |
| White n=11507 | | | | | | |
| Row % | 96.1 | 0.7 | 0.2 | 1.2 | 1.9 | 100.0 |
| Column % | 93.3 | 1.3 | 3.3 | 7.6 | 26.7 | 55.0 |
| Black n=5532 | | | | | | |
| Row % | 2.7 | 90.4 | 0.6 | 4.3 | 1.9 | 100.0 |
| Column % | 1.3 | 87.1 | 5.2 | 13.0 | 13.2 | 26.5 |
| Asian n=631 | | | | | | |
| Row % | 6.8 | 2.5 | 80.8 | 1.7 | 8.1 | 100.0 |
| Column % | 0.4 | 0.3 | 75.6 | 0.6 | 6.3 | 3.0 |
| Mixed n=2806 | | | | | | |
| Row % | 17.3 | 21.0 | 3.1 | 50.9 | 7.8 | 100.0 |
| Column % | 4.1 | 10.3 | 12.7 | 77.6 | 27.0 | 13.4 |
| Other n=440 | | | | | | |
| Row % | 27.3 | 13.4 | 5.0 | 5.2 | 49.1 | 100.0 |
| Column % | 1.0 | 1.0 | 3.3 | 1.3 | 26.7 | 2.1 |
| Total n=20916 | | | | | | |
| Row % | 56.7 | 27.5 | 3.2 | 8.8 | 3.9 | 100.0 |
| Column % | 100.0 | 100.0 | 100.0 | 100.0 | 100.0 | 100.0 |

For the purposes of this table, we have excluded pupils who were categorised as Chinese in either data source, in order to avoid disclosive cell counts.
CRIS, Clinical Record Interactive Search; NPD, National Pupil Database.

## Comparability of aggregate ethnicity

Among pupils who had usable ethnicity data from both the NPD and CRIS (n=20991), the two data sources agreed on 87.0% of categorisations, Cohen's kappa=0.79 (95% CI 0.78 to 0.79), indicating strong overall agreement.

Cross-tabulations between ethnicity data in the two data sources can be found in table 2 (note that for the purposes of this table, we have excluded pupils who were categorised as Chinese in either data source, in order to avoid disclosive cell counts). The majority of those categorised as White, Black, or Asian in the NPD were also categorised as such in CRIS (row percentages, table 2), and vice versa (column percentages, table 2). However, only 50.9% of those in the NPD's Mixed ethnic group were also classified as such in CRIS. Additionally, of those grouped as Other in CRIS, only 26.7% were also grouped as Other in the NPD; the same proportion were instead grouped as White (26.7%), and still more were grouped as Mixed (27.0%). Further detail on the proportion of disaggregated CRIS ethnicities categorised as consistent major ethnic groups in the NPD is available in online supplemental table S5.

## Association with educational and clinical outcomes

The association between ethnicity, year 11 attainment and neurodevelopmental disorder varied according to which data source was used to code ethnicity. Asian and Other ethnic groups were significantly more likely to achieve expected attainment thresholds in year 11 than White ethnic groups when ethnicity was derived from the NPD, but not when ethnicity was derived from CRIS (table 3). Also, compared with White ethnic groups, Mixed ethnic groups were less likely to receive neurodevelopmental disorder diagnoses only when derived from the NPD, and Chinese ethnic groups were less likely to receive neurodevelopmental disorder diagnoses only when derived from CRIS (although it should be noted that the cell sizes underlying the risk ratios for the Chinese ethnic group were small) (table 4). Associations with missing ethnicity data also varied according to which data source was used.

**Table 3** Unadjusted risk ratios between ethnicity derived from either NPD or CRIS (exposure) and whether the year 11 expected attainment threshold was achieved (outcome; 'no' is the reference group)

| | NPD-derived ethnicity as exposure | | | | CRIS-derived ethnicity as exposure | | | |
|---|---|---|---|---|---|---|---|---|
| | No (%) | Yes (%) | RR | 95% CI | No (%) | Yes (%) | RR | 95% CI |
| White | 4485 (67.1) | 2202 (32.9) | Reference | – | 5041 (67.2) | 2461 (32.8) | Reference | – |
| Black | 1691 (69.6) | 739 (30.4) | 0.92 | 0.86 to 0.99 | 2293 (75.1) | 761 (24.9) | 0.76 | 0.71 to 0.82 |
| Asian | 174 (52.4) | 158 (47.6) | 1.46 | 1.29 to 1.64 | 290 (63.5) | 167 (36.5) | 1.11 | 0.98 to 1.26 |
| Chinese | 15 (48.4) | 16 (51.6) | 1.59 | 1.05 to 2.13 | 17 (41.5) | 24 (58.5) | 1.78 | 1.31 to 2.20 |
| Mixed | 861 (66.3) | 438 (33.7) | 1.02 | 0.94 to 1.11 | 618 (69.1) | 276 (30.9) | 0.94 | 0.85 to 1.04 |
| Other | 110 (51.6) | 103 (48.4) | 1.48 | 1.27 to 1.70 | 369 (65.2) | 197 (34.8) | 1.06 | 0.94 to 1.19 |
| Missing | 2673 (74.8) | 902 (25.2) | 0.76 | 0.71 to 0.82 | 1381 (67.3) | 672 (32.7) | 1.00 | 0.93 to 1.07 |

n=14 567. The n=15 859 not included in these analyses were missing data on year 11 attainment.
CRIS, Clinical Record Interactive Search; NPD, National Pupil Database; RR, risk ratio.

Finally, our sensitivity analyses indicated that supplementing missing ethnicity data from one source with ethnicity data from the other source brought these risk ratios more in line with each other to some degree. However, the strength of these risk ratios still varied according to whether CRIS or the NPD was used as the primary data source (online supplemental table S6, table S7), and discrepancies in the estimated proportion of the sample belonging to each ethnic group also remained (online supplemental table S8).

## DISCUSSION

These findings suggest that linkage between health and education data sources can provide an opportunity to increase available ethnicity data for studies using linked and routinely collected records. In this sample, the proportion with available and usable ethnicity data rose from 79.3% (from the NPD) and 87.0% (from CRIS) to 97.3% (from either the NPD or from CRIS). Ethnicity at an aggregate level mostly aligned between the two data sources, with the NPD and CRIS agreeing on 87.0% of classifications. However, as with previous studies, some divergence was observed, particularly in the Mixed and Other ethnic groups.[16]

Low-quality ethnicity data are known to pose a challenge for investigating ethnic inequalities in public health.[5] A previous UK study found incomplete and inconsistent ethnicity recordings in two health datasets,[16] and outputs accessed via the UK government's online data service corroborate that collection and reporting of ethnicity data within the public sector is not consistent.[28]

This study updates and extends previous findings by comparing two health and education datasets, thereby taking a cross-disciplinary perspective on this issue. It also demonstrates that many of the same measurement issues affecting ethnicity data in other datasets and populations can be present in cross-disciplinary datasets for children and adolescents accessing mental health services. This is important because many research studies and national statistical releases make use of routinely collected ethnicity data to estimate inequalities affecting CAMHS (eg, inequalities in referral pathways) and affecting the education system (eg, inequalities in school exclusions).[29 30]

**Table 4** Unadjusted risk ratios between ethnicity derived from either NPD or CRIS (exposure) and neurodevelopmental disorder diagnosis (outcome; 'no' is the reference group)

| | NPD-derived ethnicity as exposure | | | | CRIS-derived ethnicity as exposure | | | |
|---|---|---|---|---|---|---|---|---|
| | No (%) | Yes (%) | RR | 95% CI | No (%) | Yes (%) | RR | 95% CI |
| White | 9218 (69.5) | 4041 (30.5) | Reference | – | 9963 (68.4) | 4609 (31.6) | Reference | – |
| Black | 4660 (74.2) | 1618 (25.8) | 0.84 | 0.80 to 0.89 | 5396 (72.9) | 2004 (27.1) | 0.85 | 0.81 to 0.89 |
| Asian | 562 (74.4) | 193 (25.6) | 0.84 | 0.73 to 0.95 | 716 (74.7) | 243 (25.3) | 0.79 | 0.71 to 0.89 |
| Chinese | 53 (72.6) | 20 (27.4) | 0.90 | 0.60 to 1.28 | 65 (79.3) | 17 (20.7) | 0.65 | 0.41 to 0.98 |
| Mixed | 2348 (72.3) | 899 (27.7) | 0.91 | 0.85 to 0.96 | 1577 (68.7) | 717 (31.3) | 0.99 | 0.92 to 1.06 |
| Other | 425 (82.5) | 90 (17.5) | 0.57 | 0.47 to 0.69 | 908 (77.9) | 258 (22.1) | 0.69 | 0.61 to 0.77 |
| Missing | 4590 (72.9) | 1709 (27.1) | 0.89 | 0.84 to 0.93 | 3231 (81.7) | 722 (18.3) | 0.57 | 0.53 to 0.61 |

n=30 426.
CRIS, Clinical Record Interactive Search; NPD, National Pupil Database; RR, risk ratio.

Consistent with previous work in other populations, we found that linkage between these datasets increased availability of ethnicity data, but that the datasets did not consistently agree on ethnicity, particularly for Mixed and Other ethnic groups. The Other ethnic category is known to capture particularly heterogenous groups, such that non-alignment in this category is perhaps unsurprising. Indeed, the Other classification in each source may partly comprise individuals who do not feel that their ethnicity is otherwise captured by options offered by the data collection procedure. For example, when submitting ethnicity data for the school census, a range of ethnicity options are offered by the NPD which would ultimately be aggregated as White (such as Cornish, Italian, White Eastern European and White Western European). However, these options are not available in CRIS, such that in the clinical setting the patient, parent or recording clinician might instead select 'any other group' as an alternative and be aggregated as Other. Disagreements between the data sources on the Mixed ethnic group might reflect previous UK research findings that individuals from Mixed ethnic groups are culturally positioned between White and minority ethnic groups, and how they self-identify may be particularly fluid and context dependent.[31 32]

Therefore, some inconsistencies in ethnicity between multiple data sources are expected. Ethnicity is not a time-invariant characteristic, is highly context specific, and self-reported ethnic identity has been known to vary in the same individuals over the life course.[33] However, our findings suggest that this can result in substantial differences in estimated associations between ethnicity and important clinical and educational outcomes.

Our findings pose serious challenges for accurately identifying inequalities which need to be addressed. A recent report recommended the enforcement of guidelines to ensure that patient ethnicity is recorded, and recorded accurately based on self-report, during interactions with healthcare services. Doing so might resolve some of the observed discrepancies,[5] particularly as our study indicated large differences in observed associations for children with 'missing' ethnicity. However, it is important to note that both the coverage and quality of ethnicity recording in routinely collected data need to improve; solely enforcing more comprehensive completion of ethnicity fields could cause the quality and accuracy of these data to deteriorate further if not accompanied by updated guidance on recording ethnicity.[34]

Some of the issues we identified might also be resolved by data sources supplying ethnicity data to researchers at a more granular level to ensure consistency in how ethnicity is aggregated. Understandably, concerns around data anonymity sometimes preclude this, but in light of known inequalities at the level of more granular ethnicity and growing 'super-diversity' in the UK, aggregate ethnicity data remain insufficient.[35 36] Where aggregation is essential for maintaining anonymity, other methodological improvements might include standardisation across the public sector in how ethnicity is aggregated, or

allowing researchers to specify how they want ethnicity to be aggregated when requesting data extracts. Researchers reporting analyses from linked datasets should also be transparent about how they have coded ethnicity and might consider conducting sensitivity analyses to understand the possible impact of these coding decisions on their analyses.

Efforts to effectively capture ethnicity in research are ongoing. The Office for National Statistics conducted a public consultation and stakeholder survey to inform ethnic group data collection techniques in preparation for the 2021 Census. Their findings indicated that aggregated ethnicity categories (such as White, Black, Asian, Mixed and Other) do not sufficiently meet the requirements of many data users.[37] Our findings reinforce that not only is aggregate ethnicity data insufficient, but that it does not always align between large-scale data sources. Moreover, choosing which data source to primarily code ethnicity from could have substantial implications for how associations between ethnicity, health and educational outcomes are interpreted, such that users of linked data should be cautious in combining and comparing ethnicity between different data sources.

We used broad terminology in reference to ethnicity, which is not desirable; however, as discussed, a more specific examination and discussion were precluded by only having access to aggregate ethnicity variables.[6] Additionally, the terminology follows Office for National Statistics standard approaches for aggregation which are commonly used by researchers and government analysts to inform UK policy. This is helpful for generalisability.

We only used one assessment of ethnicity per individual and per data source, and we were unable to ascertain exactly when ethnicity was ascribed. Therefore, within-person changes in reported ethnicity over time could not be assessed.

We did not adjust the risk ratios for key covariates such as socioeconomic status, gender and age. It should be emphasised that our aim was not to provide a precise estimate of the association between ethnicity and attainment or neurodevelopmental disorder, but to ascertain whether variable findings can arise when measuring ethnicity from different data sources. Moreover, gender, socioeconomic status and age were also recorded inconsistently between the datasets, so including them would raise further questions as to which data source to rely on for their classification: this is beyond the scope of this study's focus on discrepancies seen in ethnicity. However, whether these characteristics in turn predict ethnicity misclassification would be an area for future research. We also did not consider other important characteristics such as religion, first language and country of birth—future work could consider these characteristics to investigate whether linkage might inform inequalities research from an intersectional perspective.[38]

*Wickersham A, et al. BMJ Open* 2024;**14**:e078788. doi:10.1136/bmjopen-2023-078788

## CONCLUSION

Data linkage minimises missingness in ethnicity data, and ethnicity categorisations between CRIS and the NPD were mostly consistent. However, choosing which data source to primarily code ethnicity from can impact findings relating to ethnicity, mental health and education.

**Author affiliations**
[1]CAMHS Digital Lab, Department of Child and Adolescent Psychiatry, Institute of Psychiatry, Psychology & Neuroscience, King's College London, London, UK
[2]Department of Psychological Medicine, Institute of Psychiatry, Psychology & Neuroscience, King's College London, London, UK
[3]Department of Psychiatry, University of Cambridge, Cambridge, UK
[4]Maudsley Biomedical Research Centre, South London and Maudsley NHS Foundation Trust, London, UK

**Contributors** AW conceived and designed the study, performed data analysis, wrote the manuscript, and is the guarantor for the work. JD led the data linkage, conducted initial cleaning of NPD data, and provided critical comments on the manuscript. AJ performed data extraction. JD-M, TF and RS consulted on study design and interpretation, and provided critical comments on the manuscript. All authors read and approved the final manuscript. At the time of undertaking this study, JD and AW had full access to the dataset from which the study population was derived.

**Funding** This paper represents independent research funded by the National Institute for Health Research (NIHR) Biomedical Research Centre at South London and Maudsley NHS Foundation Trust and King's College London (NIHR-INF-0690). The authors also received support from elsewhere, including Medical Research Council and Economic and Social Research Council - further details on this are provided under Competing Interests.

**Disclaimer** The views expressed are those of the authors and not necessarily those of the NIHR, the Department of Health and Social Care or King's College London.

**Competing interests** AW is supported by ADR UK (Administrative Data Research UK), an Economic and Social Research Council (ESRC) investment (part of UK Research and Innovation) (ES/W002531/1). RS is part-funded by 1. the National Institute for Health Research (NIHR) Biomedical Research Centre at the South London and Maudsley NHS Foundation Trust and King's College London; 2. the National Institute for Health Research (NIHR) Applied Research Collaboration South London (NIHR ARC South London) at King's College Hospital NHS Foundation Trust; 3. the DATAMIND HDR UK Mental Health Data Hub (MRC grant MR/W014386). RS has received research support in the last 3 years from Janssen, GSK and Takeda. JD is supported by NIHR Clinician Science Fellowship award (CS-2018-18-ST2-014) and has received support from a Medical Research Council (MRC) Clinical Research Training Fellowship (MR/L017105/1) and Psychiatry Research Trust Peggy Pollak Research Fellowship in Developmental Psychiatry. JD-M is supported by the ESRC Centre for Society and Mental Health at King's College London (ESRC Reference: ES/S012567/1), grants from the ESRC (ES/S002715/1) and by the National Institute for Health Research (NIHR) Biomedical Research Centre at South London and Maudsley NHS Foundation Trust and King's College London and the National Institute for Health Research (NIHR) Applied Research Collaboration South London (NIHR ARC South London) at King's College Hospital NHS Foundation Trust. The views expressed are those of the authors and not necessarily those of the ESRC, NIHR, the Department of Health and Social Care or King's College London.

**Patient and public involvement** Patients and/or the public were involved in the design, or conduct, or reporting, or dissemination plans of this research. Refer to the Methods section for further details.

**Patient consent for publication** Not applicable.

**Ethics approval** The CRIS data resource, including the linked data used in this manuscript, has received research ethics approval for secondary analyses (Oxford REC C, reference 18/SC/0372). The CRIS Oversight Committee ensures that research conducted using health records is ethical and legal, and service users can opt-out of their CRIS data being used for research.

**Provenance and peer review** Not commissioned; externally peer reviewed.

**Data availability statement** Data may be obtained from a third party and are not publicly available. The data that support the findings of this study are available from the Department for Education and South London and Maudsley NHS Foundation Trust, but restrictions apply to the availability of these data, and so they are not publicly available. For the purposes of open access, the author has applied a Creative Commons Attribution (CC BY) licence to any Accepted Author Manuscript version arising from this submission.

**ORCID iDs**
Alice Wickersham http://orcid.org/0000-0002-7402-7690
Jayati Das-Munshi http://orcid.org/0000-0002-3913-6859
Tamsin Ford http://orcid.org/0000-0001-5295-4904
Amelia Jewell http://orcid.org/0000-0002-0887-2159
Johnny Downs http://orcid.org/0000-0002-8061-295X

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
