## [Reviewer comments · BMJ Open]

ARTICLE DETAILS

TITLE (PROVISIONAL)	The impact of inconsistent ethnicity recordings on estimates of inequality in child health and education data: A data linkage study of Child and Adolescent Mental Health services in South London
AUTHORS	Wickersham, Alice; Das-Munshi, Jayati; Ford, Tamsin; Jewell, Amelia; Stewart, Robert; Downs, Johnny

VERSION 1 – REVIEW

REVIEWER	Hanly, Mark University of New South Wales, Centre for Big Data Research in Health
REVIEW RETURNED	30-Aug-2023

GENERAL COMMENTS	Thank you for the opportunity to review your manuscript, which I read with interest. This is an important topic and valuable contributions. I have some minor suggestions below which you may wish to consider in the case of a revision. Introduction There is a fairly extensive related literature on identification of Indigenous Australians in linked administrative datasets that might be worth noting briefly in the introduction. E.g. see McNamara et al, Int J Popul Data Sci. 2020; 5(1) and references therein. Materials and Measures Would it be sensible to pool the Chinese ethnicity group with other Asian ethnicities, given the small size of the former leads to the disclosure issue in Table 2 and OR estimates with very wide confidence intervals? The small size of the Chinese ethnicity group (0.2-0.3%) relative to census-based population estimates for London (~1.5%) seems to suggest there might be some issues with this category too. The resulting groups, especially for the analysis of Year 11 outcomes, are very small. The outcomes are relatively common so I would suggest presenting risk ratios rather than odds ratio (given the OR is often interpreted as a RR but overestimates the risk when the outcome is common). About half the linked data sample were missing the outcome for the Year 11 analysis, could you briefly explain why? Is the mechanism likely to be missing completely at random (e.g. if some children haven't sat the Year 11 test or already sat it prior to the linkage window?) or are there potential selection effects that could influence the results in Figure 1 (e.g. is there differential linkage by
--

	state/public/private schools or something like that)? If the latter, perhaps you could present the proportion missing the Year 11 outcome by the NPD- and CRIS-derived ethnicity to check/confirm the outcome is at least missing at random with respect to ethnicity. Analysis If I've understood correctly, the analysis compares scenarios where ethnicity is coded from either one source or the other, leaving those with missing data as a distinct group. As you note, in practice those with missing information in one source would be supplemented with information from the alternative where possible to minimise missingness. So, is that a more practice-relevant comparison to include? i.e. NPD-derived ethnicity with missing data filled in from the CRIS (where available) versus CRIS-derived ethnicity with missing data filled in from the NPD (where available). Discussion Perhaps also emphasise practical messages for people working with multiple linked data sources e.g. (i) the importance of transparency regarding choice of coding and (ii) the importance of testing the sensitivity of results to the choice of coding when ethnicity is an key analysis variable.
--	---

REVIEWER	Powell, Madeleine UNSW
REVIEW RETURNED	26-Sep-2023

GENERAL COMMENTS	OVERALL COMMENTS This study included a population of 30,436 children aged 4-18 years who were referred to/received child and adolescent mental health services in South London and Maudsley (SLAM) NHS district between 2007-2013. The study used data collected from the SLAM e-health records and the National Pupil Database. The outcomes included neurodevelopment disorders, including intellectual disability, pervasive developmental disorder, and hyperkinetic disorder measured using primary or secondary diagnoses recorded in the e-health records, and the achievement of "expected year 11 attainment threshold" using the national pupil database. Using this specific patient population and their health and education data, the paper sets out to address two aims. The first aim is to assess the consistency of ethnicity recorded across the two data sources – that is, the e-health records and the National Pupil Database. The second aim is to examine how using different rules for defining a child's ethnicity from the available data sources changes the magnitude of the frequency of an outcome in each ethnic group, as well as the size of the inequalities in outcomes between ethnic groups. In the Introduction of the paper, the authors establish the issue of incomplete ethnicity recording in linked data sources and discusses the implications that this can have on measuring the frequency of health outcomes in the population, as well as inequalities between different ethnic groups. This is indeed an important issue for any epidemiological studies using administrative data sources to quantify the scale of health outcomes and inequalities of public health interest.
---

	My main concern with this paper, in its current form, is that the authors do not introduce the relevance of exploring this issue in a child and adolescent mental health service population, which is a very specific patient population, in the Introduction or the Discussion sections of the paper. The recording of ethnicity is a whole of population measurement bias issue, with potentially different reasons for the measurement bias in different sub-population groups, including specific health service populations (in this case, children accessing publicly funded mental health services). For example, the authors have not discussed issues around who may or may not be referred to or access publicly funded mental health services for children and adolescents, and how access to these services may vary between different ethnic groups. Below are some specific comments on different sections of the paper. Title: It would be useful to specify the specific study population in the title. Abstract It would be useful to specify the specific study population in the abstract Methods  • It would be useful to report the years of data collection, including how many years of data were available from the National Pupil Database, and from the SLAM e-health records. • It would be useful to explicitly report the different sub-groups of the overall study population included in the analyses for the three different outcomes examined under the different aims (i.e., ethnicity recording for aim one, and neurodevelopmental disorders, and year 11 attainment for aim two a, including the numbers in each and the eligibility and ages of children in each. For example, in the supplementary material it is detailed that “Year 11 attainment data” was only available for 14,567 children, and “30,426” children had the Neurodevelopmental disorder data. I assume that the children in the “Year 11” analysis were older at referral to the mental health service than the children in the “neurodevelopment analysis”, otherwise they may not be old enough to attend Year 11 (depending on how many years of data were available). Ethnicity data:  • Were ethnicity data ascertained only from the child’s mental health service e-health records? Or were all the SLAM e-health records, from mental health and other health services, available and used to ascertain the ethnicity measure? Outcomes  • It would be useful to specify what measures were used to ascertain a primary and secondary diagnosis of
--	--

	neurodevelopmental disorder recorded in the e-health records. For example, were International Classification of Diseases codes used? Analysis and results  • As this is a cohort study, the most relevant measures of association include the risk difference and the relative risk, rather than the odds ratio. It is also important for the absolute risks of the outcomes in each ethnic group to be reported. It is also important to note that using an odds ratio when the outcome of interest is common can inflate/or overestimate the difference between groups. Although not reported, I assume that year 11 attainment is fairly common, as would neurodevelopmental disorders among children referred to mental health services? • I would recommend that the authors first report how using different ethnicity recordings change the patient population profile of children accessing mental health services. For example, what is the ethnic profile of children if ethnicity data only from one data source is used compared to if both data sources are used. An exploration of the impact of ethnicity recoding inconsistencies on profiling patient populations is of interest when planning services, and public health interventions targeted toward particular population groups. Tables and figures  • In table 1, it would be useful to provide further child characteristics among the total study population, and among the different sub-groups used in the different analyses. For example, age, gender etc. • In Figures 1 and 2, it would be useful to detail the number of participants included in each analysis. • The authors should consider including tables s6 and s7 as a main result in the paper. DISCUSSION  • It would be useful to frame the discussion and implications of the results around the implications for this specific patient population, rather than whole-of-population. I wish to acknowledge that Dr Kathleen Falster provided supervision and assistance on my review of this manuscript.
--	--

VERSION 1 – AUTHOR RESPONSE

REVIEWER 1 COMMENT: Thank you for the opportunity to review your manuscript, which I read with interest. This is an important topic and valuable contributions. I have some minor suggestions below which you may wish to consider in the case of a revision.

AUTHOR RESPONSE: Many thanks for your kind and helpful feedback!

REVIEWER 1 COMMENT: There is a fairly extensive related literature on identification of Indigenous Australians in linked administrative datasets that might be worth noting briefly in the introduction. E.g. see McNamara et al, Int J Popul Data Sci. 2020; 5(1) and references therein.

AUTHOR RESPONSE: Thank you for drawing attention to this very rich literature, we have added some citations to the introduction accordingly.

REVIEWER 1 COMMENT: Would it be sensible to pool the Chinese ethnicity group with other Asian ethnicities, given the small size of the former leads to the disclosure issue in Table 2 and OR estimates with very wide confidence intervals? The small size of the Chinese ethnicity group (0.2-0.3%) relative to census-based population estimates for London (~1.5%) seems to suggest there might be some issues with this category too. The resulting groups, especially for the analysis of Year 11 outcomes, are very small.

AUTHOR RESPONSE: We are keen not to pool the Chinese ethnic group with other Asian ethnicities in order to maintain as much granularity in our ethnic groups as possible, and in order to remain consistent with groupings used by national statistical releases from the Department for Education. Regarding the difference in representation compared to population estimates, this is likely because our sample focuses on those who have been referred to secondary mental health services, rather than a general population sample (there are known to be ethnic inequalities in referral pathways to mental health services, e.g. <https://doi.org/10.1007/s00787-020-01603-7>), so does not necessarily indicate measurement issues with this category. Therefore, we have retained the Chinese ethnic group in this revision, but have included the cell frequencies in Tables 3 and 4 so that readers can take this into consideration, and have noted the small cell sizes where we refer to results arising from this group in the results section.

REVIEWER 1 COMMENT: The outcomes are relatively common so I would suggest presenting risk ratios rather than odds ratio (given the OR is often interpreted as a RR but overestimates the risk when the outcome is common).

AUTHOR RESPONSE: Thank you for this suggestion, we have presented risk ratios rather than odds ratios accordingly in Tables 3 and 4.

REVIEWER 1 COMMENT: About half the linked data sample were missing the outcome for the Year 11 analysis, could you briefly explain why? Is the mechanism likely to be missing completely at random (e.g. if some children haven't sat the Year 11 test or already sat it prior to the linkage window?) or are there potential selection effects that could influence the results in Figure 1 (e.g. is there differential linkage by state/public/private schools or something like that)? If the latter, perhaps you could present the proportion missing the Year 11 outcome by the NPD- and CRIS-derived ethnicity to check/confirm the outcome is at least missing at random with respect to ethnicity.

AUTHOR RESPONSE: The high level of missingness in our Year 11 analysis is consistent with the fact that our cohort of pupils were age 4 to 18 at the time of linking their educational records, so as the reviewer suggests, many of the children in this cohort had not yet completed their Year 11 assessments by the time of the linkage window. We have added this clarification to the methods section, and have also elaborated that Year 11 assessments are typically carried out at ages 15-16.

REVIEWER 1 COMMENT: If I've understood correctly, the analysis compares scenarios where ethnicity is coded from either one source or the other, leaving those with missing data as a distinct group. As you note, in practice those with missing information in one source would be supplemented with information from the alternative where possible to minimise missingness. So, is that a more practice-relevant comparison to include? i.e. NPD-derived ethnicity with missing data filled in from the CRIS (where available) versus CRIS-derived ethnicity with missing data filled in from the NPD (where available).

AUTHOR RESPONSE: Thank you for this suggestion, which we agree would be a practice-relevant comparison. In response to this and Reviewer 2's feedback, we have added this by way of a sensitivity analysis. Briefly, these analyses indicated that supplementing missing ethnicity data from one source with ethnicity data from the other source brought the risk ratios more in line with each other to some degree. However, the strength of these risk ratios still varied according to whether CRIS or the NPD was used as the primary data source (Table S6, Table S7), and discrepancies in the estimated proportion of the sample belonging to each ethnic group also remained (Table S8).

REVIEWER 1 COMMENT: Perhaps also emphasise practical messages for people working with multiple linked data sources e.g. (i) the importance of transparency regarding choice of coding and (ii) the importance of testing the sensitivity of results to the choice of coding when ethnicity is a key analysis variable.

AUTHOR RESPONSE: These are excellent points, and we have emphasised them further in our discussion section.

REVIEWER 2 COMMENT: This study included a population of 30,436 children aged 4-18 years who were referred to/received child and adolescent mental health services in South London and Maudsley (SLAM) NHS district between 2007-2013. The study used data collected from the SLAM e-health records and the National Pupil Database. The outcomes included neurodevelopment disorders, including intellectual disability, pervasive developmental disorder, and hyperkinetic disorder measured using primary or secondary diagnoses recorded in the e-health records, and the achievement of "expected year 11 attainment threshold" using the national pupil database.

Using this specific patient population and their health and education data, the paper sets out to address two aims. The first aim is to assess the consistency of ethnicity recorded across the two data sources – that is, the e-health records and the National Pupil Database. The second aim is to examine how using different rules for defining a child's ethnicity from the available data sources changes the magnitude of the frequency of an outcome in each ethnic group, as well as the size of the inequalities in outcomes between ethnic groups. In the Introduction of the paper, the authors establish the issue of incomplete ethnicity recording in linked data sources and discusses the implications that this can have on measuring the frequency of health outcomes in the population, as well as inequalities between different ethnic groups. This is indeed an important issue for any epidemiological studies using administrative data sources to quantify the scale of health outcomes and inequalities of public health interest.

AUTHOR RESPONSE: Many thanks for your kind and thorough feedback!

REVIEWER 2 COMMENT: My main concern with this paper, in its current form, is that the authors do not introduce the relevance of exploring this issue in a child and adolescent mental health service population, which is a very specific patient population, in the Introduction or the Discussion sections of the paper. The recording of ethnicity is a whole of population measurement bias issue, with potentially different reasons for the measurement bias in different sub-population groups, including specific health service populations (in this case, children accessing publicly funded mental health services). For example, the authors have not discussed issues around who may or may not be referred to or access publicly funded mental health services for children and adolescents, and how access to these services may vary between different ethnic groups.

AUTHOR RESPONSE: Thank you for these reflections. The potentially varying measurement bias issues which the reviewer notes are indeed one of the key motivations for exploring this issue in a child and adolescent mental health service population. Ethnic biases in referral patterns to CAMHS have been the subject of extensive research elsewhere (e.g. <https://doi.org/10.1007/s00787-020-01603-7>), and while our analyses cannot speak specifically to this issue, we agree that issues around

measurement bias affect our ability to unpick these other biases affecting CAMHS and educational services, which is why understanding this measurement bias in this specific population is so important. We have clarified this position towards the end of our introduction section.

However, our finding that many of the same biases identified in other populations were identified in our child and adolescent mental health service population shows that our findings are indeed consistent with and reflective of many of the same issues affecting other datasets and other populations. This is why our discussion section widens beyond the specific population under study. Again, we have now clarified this position in our discussion section.

REVIEWER 2 COMMENT: Title: It would be useful to specify the specific study population in the title.

AUTHOR RESPONSE: We have clarified in the title that the study focuses on Child and Adolescent Mental Health Services in South London.

REVIEWER 2 COMMENT: Abstract: It would be useful to specify the specific study population in the abstract

AUTHOR RESPONSE: We have clarified in the abstract that the study focuses on Child and Adolescent Mental Health Services (the location had already been specified).

REVIEWER 2 COMMENT: Methods: • It would be useful to report the years of data collection, including how many years of data were available from the National Pupil Database, and from the SLAM e-health records.

AUTHOR RESPONSE: The years covered by the data linkage, and the date of data extraction from CRIS, are provided in the methods section. Beyond this, we do not have more granular data on the number of years of NPD data available, or when specific variables were gathered (both of which will vary for each pupil). We acknowledge this specifically in relation to ethnicity in our limitations section. This is also the rationale for why we focus on neurodevelopmental disorders, which as explained in the methods section, are often pervasive and lifelong, therefore overcoming biases which might arise in differential timing of diagnoses.

REVIEWER 2 COMMENT: Methods: • It would be useful to explicitly report the different sub-groups of the overall study population included in the analyses for the three different outcomes examined under the different aims (i.e., ethnicity recording for aim one, and neurodevelopmental disorders, and year 11 attainment for aim two a, including the numbers in each and the eligibility and ages of children in each. For example, in the supplementary material it is detailed that “Year 11 attainment data” was only available for 14,567 children, and “30,426” children had the Neurodevelopmental disorder data. I assume that the children in the “Year 11” analysis were older at referral to the mental health service than the children in the “neurodevelopment analysis”, otherwise they may not be old enough to attend Year 11 (depending on how many years of data were available).

AUTHOR RESPONSE: Thank you for this feedback. Tables 1 and 2 contain ethnicity information relevant to Aims 1 and 2. We have also now added Tables 3 and 4 into the main manuscript, which contains the frequencies of the different ethnic groups stratified by the two different outcomes used in Aim 3 (neurodevelopmental disorder and Year 11 attainment).

In terms of the eligibility and ages of children, all children in the linkage were eligible to be included in all of our aims, which is why the flow diagram in Figure S1 focuses primarily on the availability of ethnicity and outcome data, rather than eligibility. We have added further detail to Figure S1 for clarity (note that because the ‘missing’ ethnicity category was included in regressions, the sample sizes for

these regressions were not restricted by ethnicity data availability – hence this forms a separate part of the flow diagram). As explained above, the ages and years that ethnicity and neurodevelopmental disorder were routinely collected will vary for each child, and more granular information on this is not available in our dataset. Year 11 assessments are typically carried out at ages 15-16, which we have now clarified in the methods section.

The high level of missingness in our Year 11 analysis is consistent with the fact that our cohort of pupils were age 4 to 18 at the time of linking their educational records, so many of the children in this cohort had not yet completed their Year 11 assessments by the time of the linkage window. We have also added this clarification to the methods section.

REVIEWER 2 COMMENT: Methods: Ethnicity data: • Were ethnicity data ascertained only from the child's mental health service e-health records? Or were all the SLAM e-health records, from mental health and other health services, available and used to ascertain the ethnicity measure?

AUTHOR RESPONSE: CRIS draws information from all South London and Maudsley NHS Foundation Trust (SLaM) clinical records. So, a patient's ethnicity may be recorded during their interactions with any SLaM service (including, but not limited to, CAMHS). In other words, if a child had also been seen by another SLaM service in addition to CAMHS, it is possible that their ethnicity field was populated during their interaction with that service, rather than during their interaction with CAMHS specifically. We have added this clarification to the methods section.

REVIEWER 2 COMMENT: Methods: Outcomes: • It would be useful to specify what measures were used to ascertain a primary and secondary diagnosis of neurodevelopmental disorder recorded in the e-health records. For example, were International Classification of Diseases codes used?

AUTHOR RESPONSE: We have added this detail to the methods section.

REVIEWER 2 COMMENT: Methods: Analysis and results: • As this is a cohort study, the most relevant measures of association include the risk difference and the relative risk, rather than the odds ratio. It is also important for the absolute risks of the outcomes in each ethnic group to be reported. It is also important to note that using an odds ratio when the outcome of interest is common can inflate/or overestimate the difference between groups. Although not reported, I assume that year 11 attainment is fairly common, as would neurodevelopmental disorders among children referred to mental health services?

AUTHOR RESPONSE: Thank you for this suggestion, we have revised our manuscript to report risk ratios rather than odds ratios in response to this and Reviewer 1's feedback, Tables 3 and 4. The frequencies of the outcomes were reported in Table 1.

REVIEWER 2 COMMENT: Methods: Analysis and results: • I would recommend that the authors first report how using different ethnicity recordings change the patient population profile of children accessing mental health services. For example, what is the ethnic profile of children if ethnicity data only from one data source is used compared to if both data sources are used. An exploration of the impact of ethnicity recoding inconsistencies on profiling patient populations is of interest when planning services, and public health interventions targeted toward particular population groups.

AUTHOR RESPONSE: Thank you for this suggestion. The ethnicity profile of the sample was shown to vary depending on the data source used in Table 1, but as suggested we have now also added Table S8, which shows the same information if missing ethnicity from one data source is supplemented with ethnicity from the other data source. We have presented the information here in

the supplement, rather than at the start of the results section, because it forms one part of a wider sensitivity analysis requested by Reviewer 1.

REVIEWER 2 COMMENT: Tables and figures: • In table 1, it would be useful to provide further child characteristics among the total study population, and among the different sub-groups used in the different analyses. For example, age, gender etc.

AUTHOR RESPONSE: The reason we had chosen not to include other sociodemographic data like age and gender in Table 1 is because each of those variables raise equally challenging questions about which dataset those data should be derived from, as like ethnicity they are available from both CRIS and the NPD, and can vary between the two. Unpicking this is beyond the scope of this study's very specific focus on ethnicity, but we do recognise that readers will likely want this information – therefore, we have now added a citation in the results section to a previous study where readers can find further sociodemographic information for the linked cohort.

REVIEWER 2 COMMENT: Tables and figures: • In Figures 1 and 2, it would be useful to detail the number of participants included in each analysis.

AUTHOR RESPONSE: Because Tables 3 and 4 have now been moved to the main manuscript in accordance with the reviewer's below feedback, we have removed Figures 1 and 2 to avoid unnecessary duplication of findings in multiple formats.

REVIEWER 2 COMMENT: Tables and figures: • The authors should consider including tables s6 and s7 as a main result in the paper.

AUTHOR RESPONSE: As suggested, we have moved these tables to the main manuscript, and they are now Tables 3 and 4.

REVIEWER 2 COMMENT: Discussion: • It would be useful to frame the discussion and implications of the results around the implications for this specific patient population, rather than whole-of-population.

AUTHOR RESPONSE: We have added some discussion more specific to this patient population as suggested. As we have noted above, our finding that many of the same biases identified in other populations were identified in our child and adolescent mental health service population shows that our findings are indeed consistent with and reflective of many of the same issues affecting other datasets and other populations. This is why our discussion section widens beyond the specific population under study. We have now clarified this position in our discussion section.

VERSION 2 – REVIEW

REVIEWER	Hanly, Mark University of New South Wales, Centre for Big Data Research in Health
REVIEW RETURNED	07-Dec-2023
GENERAL COMMENTS	Thank you for your detailed response to the queries raised in the review. One final comment is that the reference to logistic regression in the abstract possibly needs to be updated as appropriate, given the switch from ORs to RRs.

REVIEWER	Powell, Madeleine UNSW
REVIEW RETURNED	19-Dec-2023

GENERAL COMMENTS	The authors have made a conscientious effort to incorporate the suggested changes. They have addressed all feedback and the changes align well with the recommendations and enhance the clarity and robustness of the content.
--